# Performance of Graph and Relational Databases in Complex Queries

Petri Kotiranta, Marko Junkkari * and Jyrki Nummenmaa

Faculty of Information Technology and Communication Sciences, Computing Sciences, Tampere University, 33100 Tampere, Finland; petri.kotiranta@tuni.fi (P.K.); jyrki.nummenmaa@tuni.fi (J.N.)
* Correspondence: marko.junkkari@tuni.fi

**Abstract:** In developing NoSQL databases, a major motivation is to achieve better efficient query performance compared with relational databases. The graph database is a NoSQL paradigm where navigation is based on links instead of joining tables. Links can be implemented as pointers, and following a pointer is a constant time operation, whereas joining tables is more complicated and slower, even in the presence of foreign keys. Therefore, link-based navigation has been seen as a more efficient query approach than using join operations on tables. Existing studies strongly support this assumption. However, query complexity has received less attention. For example, in enterprise information systems, queries are usually complex so data need to be collected from several tables or by traversing paths of graph nodes of different types. In the present study, we compared the query performance of a graph-based database system (Neo4j) and relational database systems (MySQL and MariaDB). The effect of different efficiency issues (e.g., indexing and optimization) were included in the comparison in order to investigate the most efficient solutions for different query types. The outcome is that although Neo4j is more efficient for simple queries, MariaDB is essentially more efficient when the complexity of queries increases. The study also highlighted how dramatically the efficiency of relational database has grown during the last decade.

**Keywords:** graph database; relational database; performance; complex queries; Neo4J; MariaDB; MySQL

## 1. Introduction

Performance is one of the motivations to use NoSQL databases instead of traditional SQL databases [1,2]. With data and queries suitable for the data model, NoSQL databases might offer significant performance benefits. In the present study, we compared database systems of the traditional relational model and of the NoSQL graph model. In the graph model [3], which is one of the four major NoSQL types, the data consist of nodes and edges, and it has its own benefits when handling relationship rich data. While in SQL databases multiple tables may need to be joined for a relational query, in graph databases relational information can be queried by navigating through the graph.

Previous studies [4–8] where the performance of graph databases, especially Neo4j, was compared with the traditional SQL databases, indicate that graph databases possess better performance than relational databases. However, those studies mainly focused on quite simple queries. In contrast to the earlier studies, we investigated the performance of database systems in situations where the query complexity increased. In a complex query, the necessary data must be collected from several tables in an SQL database, or by traversing a path of different types of nodes, potentially using recursion, in a graph database. Using a complex query, an aggregated value (e.g., a count or an average) from a large data set can be calculated. Complex queries are typical in various application domains such as Enterprise information systems [9], Geographical information systems [10], Bioinformatics [11] and CAD systems [12]. The sample data of the present paper relate to Enterprise information systems.

In the present study, MariaDB and two versions of MySQL were selected as relational database systems. MariaDB was selected because it is a modern database system and, to the best of our knowledge, it has not been compared to graph database systems before. MySQL 8.0 was included in the present investigation in order to compare how the query performance of complex queries differs between MySQL and MariaDB. Old MySQL 5.1 was included to observe the efficiency development of relational database systems. MariaDB and MySQL 8.0 were initially based on MySQL 5.1. Thus, they belong to the same database system family, and their comparison is an indication of how relational database query efficiency has developed during the last decade. In addition to using complex queries, we also paid attention to factors related to efficiency. Indexing is a traditional method to improve performance and it can be applied with both relational and graph databases. Additionally, for the selected graph database system (Neo4J), a more efficient query execution type (call-function) was developed. Recursive queries can be optimized in modern versions of Neo4J. We considered all these optimizations in the efficiency evaluations.

In order to benchmark the database systems using complex queries, we designed and implemented a new test bench that also supports complex queries, unlike existing benchmarks such as [13] or [14]. Our test bench was designed for testing queries using MariaDB, MySQL, and Neo4J. The test database relates to enterprise information systems, but it is worth noting that the query types are general, and processing of the data is similar in many other domains. The test bench is called Invoicing Database Test Bench and its source code is available in GitHub [15]. The program generated a selected amount of data for our test invoicing database schema and performed various query tests. Our dataset is public. The source code for generating the data is available in GitHub, and, thus, it is possible for anyone to repeat these tests by installing the same test bench and generating the same data.

The rest of the paper is organized as follows. Section 2 reviews previous work related to Neo4j and MariaDB performance analysis. Section 3 introduces the schema that is used for the test data. Section 4 presents the implemented benchmarking program. Section 5 presents the test queries. Section 6 presents the test results. Section 7 discusses the results and Section 8 contains the conclusions.

## 2. Related Work

An older MySQL version was included in the comparison to make our research compatible with earlier studies. From this perspective, MariaDB is a natural choice for a modern database, because MariaDB was initially a descendant of MySQL. Based on the popularity of databases, the DB-Engines site ranks MariaDB as 8th out of 138 of relational databases [16]. Neo4j ranks 1st out of 32 graph databases on the same sites. DB-Engines ranks the databases according to current popularity. Popularity is measured using six parameters. The first parameter is the number of mentions on the websites Google and Bing. Second is the general interest which is measured by frequency in Google Trends. Third is the frequency of technical questions in Stack Overflow or DBA Stack Exchange. Fourth is the number of job offers in Indeed and Simply Hired. Fifth is the number of profiles in LinkedIn in which the system is mentioned. Sixth is the relevance in social networks which is counted by the number of tweets on Twitter, in which the system is mentioned. As all of the databases we studied are quite popular and are often candidates for use in enterprises. One of the goals of this study was to identify differences in what use case the databases should be used.

SQL databases and Neo4j have been compared in several studies [4–8]. Khan et al. compared tuned Oracle 11 g and Neo4j 3.03 Community Edition [4]. They used healthcare data, including data of patients, medication, and medical staff. Performance of the databases was evaluated using ten different count(*) queries. Many of the queries included some table joins. A physical database tuning technique called tablespaces was used for Oracle. The same databases were compared without physical database tuning by Khan et al. [6]. The physical database tuning technique decreased the overall average

query time of Oracle from 4.34 to 2.78 s. However, the overall average query time for Neo4j in query tests was only 0.67 s. Thus, Neo4j outperformed Oracle.

Holzschuher et al. tested Neo4j version 1.8 performance with different backend solutions [5]. Neo4j was benchmarked as embedded with native object access, as a dedicated server through RESTful Web Services, with embedded Cypher queries, with Cypher optimized for remote execution with REST, and with Gremlin queries through REST. MySQL version 5.5.27 was also included with Java Persistence API based backend. Queries were written using Cypher, Gremlin and SQL query languages. The test data consisted of data of persons and their relationships. Tests included such queries as friends of friends. As the size of the database increased, the advantages of Neo4j over MySQL became more evident. Neo4j performance stayed nearly constant when MySQL performance dropped by factors of 5 and 7–9. Queries in Neo4j query languages Gremlin and Cypher executed faster than queries using MySQL with JPA.

Vicknair et al. compared MySQL Community Server version 5.1.42 and Neo4j version 1.0-b11 in 2010 [7]. The graph database was transferred into a relational database as nodes and edges. Three types of structural and three types of data queries were made. The first structural query found all orphan nodes and the two other structural queries traversed the graph at depths of 4 and 128. The data queries were count(*) queries counting nodes with certain payloads. Neo4j performed better in structural queries. However, in data queries, MySQL was more efficient, partly due to the use of Lucene indexing in the tested Neo4j. The data contained integers, and Lucene treated the data as text by default, so conversions were necessary and thus impacted the performance. The work [7] by Vicknair et al. has been referenced in [4–6].

Batra et al. compared MySQL version 5.1.41 and Neo4j Community version 1.6 in 2012 [8]. They used a schema with tables user, friends, fav_movies, and actors for testing, and they tested the databases with three queries: "Find all friends of Esha", "Find all favourite movies of Esha's friends" and "Find the lead actors of Esha's friends' favourite movies". Queries were executed on 100 and 500 objects. Neo4j had 2–5 times faster query execution times with a 100-objects data set and 15–30 times faster query execution with a 500-objects data set. The work by Batra et al. [8] was similar to that of the present study as the data were stored in an SQL database with a relational schema unlike in the work by Vicknair et al. [7]. The work [8] by Batra et al. is referenced in [5].

There also exist previous performance studies where MariaDB is involved. Tongkaw et al. compared the performance of MariaDB 10.0.21 and MySQL 5.6 [17]. They used the Sysbench and OLTP [14] software systems with OLTP-Simple and OLTP-Seats workloads. Both databases consumed the same number of resources. However, when increasing the number of threads in OLTP-Simple and the number of workers in OLTP-Seats, MySQL became clearly more efficient and outperformed MariaDB. Shalygina et al. studied the Common Table Expression capabilities of MariaDB by comparing it to Postgres [18]. The study showed that Postgres had better results, when only a few steps of recursion were needed. However, MariaDB was a better choice for longer-executing recursive queries on huge amounts of data.

Stanescu [19] compared the performance of SQL Server 2009 and Neo4j 4.0. Four datasets were used consisting of 350,000, 700,000, 1,400,000 and 2,100,000 entries. A schema with multiple relations between entities was used. Five different query tests were performed with the different datasets. All the queries addressed relations between entities, so joins or matches between relationships were used. The results show that as query complexity and dataset complexity grew, Neo4j performed faster than the SQL Server.

Sholichah et al. [20] compared MySQL and Neo4j. Four queries were used. Three of the queries were tested with datasets containing 10, 100, 500, 1000 and 10,000 records. The fourth query was used to infer the databases' ability to handle unstructured data. In these tests, MySQL was in general faster and used less memory as the query complexity and number of records increased. Both databases were able to handle unstructured data.

Cheng et al. [21] compared RocksDB 5.8, Hbase 2.2, Cassandra 3.11, Neo4j 3.4.6 and MySQL 5.7. Four relational datasets from TPC-H benchmark were used as well as four real graph datasets. Different types of query workloads were tested, including atomic relational queries such as projection, aggregation, join and order by, TPC-H workloads, and different graph query algorithms. The conclusion of the tests was that relational databases outperformed graph databases with workloads that mainly consisted of group by, sort, aggregation operations and their combinations. Graph databases outperformed relational databases with workloads that mainly consisted of multiple table joins, pattern matching, path identification and their combinations.

## 3. Test Database

Our test database is a general example from enterprise information systems, where different types of information are associated with customers. In the example, a customer may have several targets for which different types of works and items are associated. When sending an invoice to a customer, the stored works and items must be found and calculated which may require complex navigation among stored data. As it is an invoicing database, one of the most important use cases is the calculation of the price for a customer invoice. This is achieved by calculating the used time for work of different work types and the price of the items used when working. Invoices might also have relations to other invoices if several invoices are sent to the customer.

The relational database has 10 tables. The basic tables customer, invoice, target, work, worktype and item represent entities of the application domain. These tables contain the customer information, customer's invoices, the target (or project) where the work is performed, a listing of each work, and a listing of different worktypes with different prices and information about the items used for each work. Relationships between the entities are stored in relationship tables of worktarget, workinvoice, useditem and workhours. These represent many-to-many relationships between entities. Figure 1 shows the database structure as a relational database schema. Arrows illustrate how the tables are associated with each other. For example, the arrow from the invoice to the customer means that customer_id in the invoice table refers to an id in the customer table.

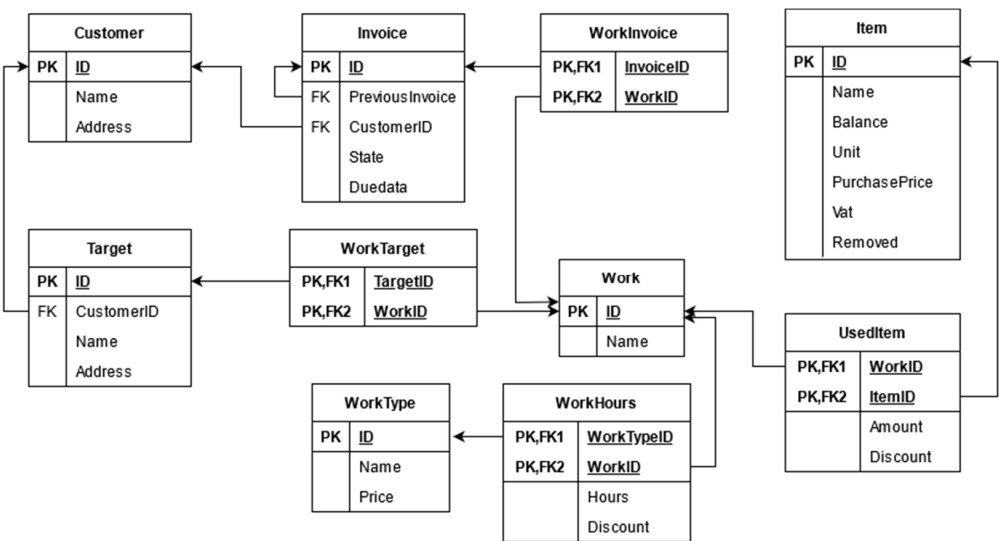

**Figure 1.** Database structure in relational format.

In our graph database schema, entities are represented as nodes, and relationships as directed edges. Two edges are used to represent many-to-many relationships. Customer, invoice, target, work, and worktype entities are represented as nodes. Relationships PAYS from the customer to invoice, and CUSTOMER_TARGET from the customer to the target, and PREVIOUS_INVOICE from an invoice to another are represented by directed edges,

the last being a recursive relationship. WORK_TARGET, WORK_INVOICE, WORKHOURS and USED_ITEM are each represented by two edges. Figure 2 represents the database structure in a graph format. The attributes of nodes and edges are not illustrated.

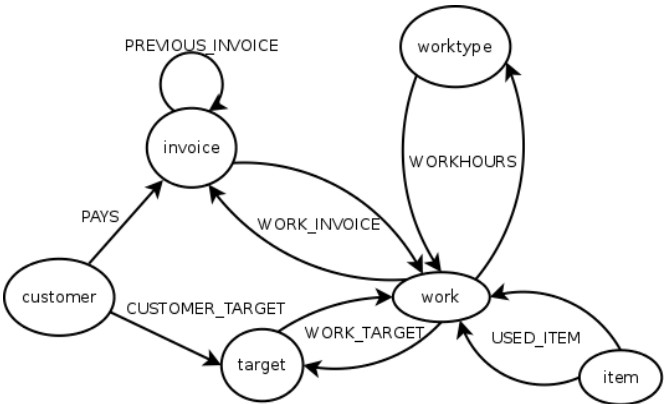

**Figure 2.** Graph database structure.

## 4. Test Program

The test data were generated using a Java program. The customer and target generation used sample data based on openly available name and address data sets [22,23]. The generation process was divided into three parts. Items and work types need to be generated first, then work and customer data. The program has threaded classes for each part. Multiple threads can be used to insert the generated data. For random data generation, controlled random seeds were used, making the generation repeatable.

The generation was controlled with parameters for the following: numbers of work types; numbers of items; numbers of related invoices, targets and work for a customer; number of works; number of customers; numbers of relations between worktypes and works; numbers of invoices and targets for each customer; and numbers of workinvoice and worktarget relationships.

The program has a class called QueryTester that was used to perform the query tests. Query tests were repeated the selected number of times. The test program collects the performance figures from the executions into a list structure. The program removes the biggest and the smallest number from the list and calculates the average and the standard deviation from the rest of the results.

The test data follows the schemata of the databases given in Figures 1 and 2. The used dataset was generated using the test program [15]. Table 1 shows the numbers of rows/objects generated for the dataset. For each row in the relationship tables of useditem, workhours, workinvoice and worktarget, two respective edges were generated for the Neo4j graph database, as a many-to-many relationship was expressed as a bidirectional relationship, i.e., two edges. The size of relational databases is 214 Mt and the size of Neo4J is 1.12 Gt.

**Table 1.** The numbers of the generated rows/objects in SQL and Neo4j.

| Table/Object | Rows in SQL | Object in Neo4J |
|---|---|---|
| Customer | 10,000 | 10,000 nodes |
| Invoice | 100,000 | 100,000 nodes |
| Item | 100,000 | 100,000 nodes |
| Target | 100,000 | 100,000 nodes |
| Work | 10,000 | 10,000 nodes |
| Workhours | 100,000 | 200,000 edges |
| Workinvoice | 1,000,000 | 2000,000 edges |
| Worktarget | 1,000,000 | 2,000,000 edges |
| Worktype | 100,000 | 100,000 nodes |
| UsedItem | 100,000 | 200,000 edges |
| Pays | - | 100,000 edges |
| Customertarget | - | 100,000 edges |
| Previousinvoice | 100/1000 | 100/1000 edges |

## 5. Test Queries

The query tests contain queries with different complexities. A query task represents an information need to be fulfilled using a query to the database, and it is implemented in SQL and Cypher queries. Each task involves the following two Cypher queries: basic form and optimized/CALL forms. The query tasks are ordered from simple to complex starting from the work price and the work price with items ending in the invoice prices, and invoice prices for a given customer. Finally, recursive queries combine all the related invoices.

The tasks were chosen as they represent typical information needs that would be executed in the chosen test databases. Finding and calculating invoice related information the primary use for a database, and this is what all the test queries demonstrated. Querying all the information required for invoices leads to complex queries. Simpler queries were included in order to see how databases perform with different complexities of queries.

Calculating the invoice prices is one of the most important query tasks. The schema does not store invoice prices explicitly. The price must be calculated based on the amount of workhours and the items used. The "price of work" and the "price of work with items" are the subqueries for calculating this price. The queries calculating invoice prices for a given customer add customer information into this task. The recursive queries find all the recursively related invoices given the top-level invoice.

### 5.1. Query Optimization

In addition to just testing databases with the queries, we investigated improving the query performance, in particular by indexing. Both MySQL and MariaDB create an index by default for the primary keys and for the foreign keys while Neo4j does not create indices for properties by default. The effect of indexing is also different in an SQL database and in a graph database. As was the case for our investigation, in SQL databases, the tables are often joined on primary key and foreign key information. Thus, SQL databases usually benefit from the indexing of primary keys and foreign keys. In a graph database, the graph is traversed when querying data. We do not benefit from indexing the properties the way we do with SQL databases. However, starting points for traversing the graph may be found faster using an index.

In order to study the effects of indexing, certain columns and properties that were used in queries were indexed in all the databases. Table 2 shows the extra indices created. As ids in customer and invoice tables are indexed by default in MySQL and MariaDB, an extra index was not needed.

Besides indexing, in Neo4j 4.4.8, queries can be optimized using CALL subqueries [24]. The CALL clause makes it possible to execute subqueries in other queries. It is similar to a function that obtains input parameters from the main query and returns some values. The subquery is executed for each incoming input row from calling the query from the main query. CALL was supported from Neo4j 4.1 onwards. In the present study, Cypher queries with and without CALL were used for backward compatibility and to see how much CALL subqueries improve the query performance.

**Table 2.** Indexed columns/properties in SQL and Neo4j.

| Table/Node | SQL | Neo4j |
|------------|-----|-------|
| Customer | - | customerId |
| Invoice | previousinvoice | invoiceId, previousinvoice |
| Item | purchaseprice | purchaseprice |
| Workhours | hours, discount | hours, discount |
| Worktype | price | price |
| Useditem | amount, discount | amount, discount |

### 5.2. Task 1: Price of Work

The first query task focused on calculating the price of works. One work can have different work types with different prices. The price of one work is defined by the number

of hours performed of the work type. There can also be a discount on the prices and the discount is included in the calculation. The related queries show how databases perform with a fairly simple query task. Figure 3 shows the queries of the task for the price of work in SQL and Cypher.

| SQL (Task 1) |
|---|

```
SELECT work.id AS workId, SUM((worktype.price * workhours.hours *
   workhours.discount))
AS price
FROM work INNER JOIN workhours ON work.id = workhours.workId
   INNER JOIN worktype
   ON worktype.id = workhours.worktypeId
GROUP BY work.id
```

| Cypher (Task 1) |
|---|

```
MATCH (wt:worktype)-[h:WORKHOURS]->(w:work)
   WITH SUM(h.hours*h.discount*wt.price)AS price, w
RETURN w.workId as workId, price
```

| Cypher with CALL (Task 1) |
|---|

```
TCH (w:work) CALL {
   WITH w
   MATCH (wt:worktype)-[h:WORKHOURS]->(w)
   RETURN SUM((h.hours*h.discount*wt.price))AS price}
RETURN w.workId as workId, price
```

**Figure 3.** Queries for Task 1 (Price of work).

*5.3. Task 2: Price of Work with Items*

The task for calculating the price of work with items is an extended version of Task 1. During the task, item prices are added into work prices. As items are also included, longer queries are needed. With this task, it is possible to see how databases perform when more relationships and calculations are included in queries. Item purchase price is a floating-point number so this changes the calculations slightly. Figure 4 shows the queries for the price of work with items in SQL and Cypher.

| SQL (Task 2) |
|---|

```
SELECT work.id AS workId, SUM((worktype.price * workhours.hours *
   workhours.discount) + (item.purchaseprice * useditem.amount *
   useditem.discount)) AS price
FROM work INNER JOIN workhours
  ON work.id = workhours.workId
   INNER JOIN worktype
   ON worktype.id = workhours.worktypeId
   INNER JOIN useditem
  ON work.id = useditem.workId
   INNER JOIN item
  ON useditem.itemId = item.id
GROUP BY work.id
```

| Cypher (Task 2) |
|---|

```
MATCH (wt:worktype)-[h:WORKHOURS]->(w:work) -[u:USED ITEM]->(i:item)
   WITH SUM((h.hours*h.discount*wt.price)+
     (u.amount*u.discount*i.purchaseprice)) AS price, w
RETURN w.workId as workId, price
```

| Cypher with CALL (Task 2) |
|---|

```
MATCH (w:work)
CALL {
   WITH w
   MATCH (wt:worktype)-[h:WORKHOURS]->(w)- [u:USED ITEM]->(i:item)
RETURN
   SUM((h.hours*h.discount*wt.price) + (u.amount*u.discount*i.purchaseprice))
   AS price }
RETURN w.workId as workId, price
```

**Figure 4.** Queries for Task 2 (Price of work with items).

*5.4. Task 3: Invoice Price*

Queries related to Task 3 calculate the sum of work prices for an invoice. The queries contain two subqueries. The first one finds the relationships between invoices and work. The second is the one presented above in Task 2. The results of these queries are joined and the sums of prices are aggregated based on the id of the invoice. This is one of the most demanding tasks and as such it is useful to see the performance differences when executing complex queries. Figure 5 shows the queries for calculating the invoice price in SQL and Cypher.

| SQL (Task 3) |
|---|

```
SELECT q1.invoiceId, SUM(q2.price) AS invoicePrice
FROM (
  SELECT workinvoice.invoiceId, workinvoice.workId
  FROM workinvoice
    INNER JOIN invoice ON workinvoice.invoiceId = invoice.id)
  AS q1 INNER JOIN (
    SELECT workhours.workid AS workId,
      SUM((worktype.price * workhours.hours * workhours.discount) +
        (item.purchaseprice * useditem.amount * useditem.discount))AS price
    FROM workhours INNER JOIN worktype ON workhours.worktypeid = worktype.id
      INNER JOIN useditem ON workhours.workid = useditem.workid INNER JOIN item
      ON useditem.itemid = item.id
  GROUP BY workhours.workid)
  AS q2 USING (workId)
GROUP BY q1.invoiceId
```

| Cypher (Task 3) |
|---|

```
MATCH (inv:invoice)-[:WORK INVOICE]->(w:work)
  WITH inv, w
  OPTIONAL MATCH (wt:worktype)-[h:WORKHOURS] ->(w:work)-[u:USED ITEM]->(i:item)
  WITH inv, w, SUM((h.hours*h.discount*wt.price) +
    (u.amount*u.discount*i.purchaseprice)) AS workPrice
RETURN inv, SUM(workPrice) as invoicePrice
```

| Cypher with CALL (Task 3) |
|---|

```
CALL {
  WITH inv
  MATCH (inv)-[:WORK INVOICE]->(w:work)
  RETURN w }
CALL {
  WITH w
  MATCH (wt:worktype)-[h:WORKHOURS]->(w)-[u:USED ITEM]->(i:item)
RETURN SUM((h.hours*h.discount*wt.price)+(u.amount*u.discount*i.purchaseprice))
  AS workPrice }
RETURN inv, SUM(workPrice) AS invoicePrice
```

**Figure 5.** Queries for Task 3 (Invoice price).

*5.5. Task 4: Invoice Prices for a Given Customer*

It is often necessary to find out all the invoice prices for a given customer. The queries of this task calculate invoice prices for a given customer. They are extensions of the queries of Task 3. Subqueries to obtain relationships between the underlying customer and invoices are included. The queries of this task are the most complex of the tested queries. From the technical point of view, the queries show how databases perform when there is a certain key defined for which the data should be related to. Figure 6 represents the queries for calculating invoice prices for a given customer.

*5.6. Task 5: Recursive Queries for Invoice Chain*

The recursive queries find all the invoices related to a given invoice id. The task is useful to test the recursive query capabilities of the databases. In SQL, Common Table Expressions are used to make the query. In Cypher, there is a way to optimize a recursive query by negating irrelevant relationships. The optimized query does not return exactly the same result as the basic query. While the basic query returns a set of individual nodes, the optimized query returns a list structure containing nodes. However, it still returns similar

results and as such it is a relevant query. Figure 7 presents the queries for finding invoices recursively related for a given invoice.

```
                              SQL (Task 4)

SELECT q1.customerId, q2.invoiceId, SUM(q3.price) AS invoicePrice
  FROM (SELECT customer.id AS customerId, invoice.id AS invoiceId
    FROM invoice INNER JOIN customer ON invoice.customerId=customer.id) AS q1
      INNER JOIN (
       SELECT workinvoice.invoiceId, workinvoice.workId
       FROM workinvoice INNER JOIN invoice
         ON workinvoice.invoiceId = invoice.id) AS q2 USING (invoiceId)
           INNER JOIN (
             SELECT workhours.workid AS workId, SUM((worktype.price * workhours.hours
             * workhours.discount) + (item.purchaseprice * useditem.amount *
             useditem.discount)) AS price
             FROM workhours
             INNER JOIN worktype ON workhours.worktypeid = worktype.id
             INNER JOIN useditem ON workhours.workid = useditem.workid
             INNER JOIN item ON useditem.itemid = item.id
           GROUP BY workhours.workid) AS q3 USING (workId)
       WHERE q1.customerId=0
       GROUP BY q2.invoiceId
```

```
                             Cypher (Task 4)

MATCH (c:customer)-[:PAYS]->(inv:invoice) WHERE c.customerId=0
  WITH c, inv
OPTIONAL MATCH (inv)-[:WORK INVOICE]->(w:work)
  WITH c, inv, w
OPTIONAL MATCH (wt:worktype)-[h:WORKHOURS]->(w:work)-[u:USED ITEM]->(i:item)
  WITH c, inv, w,
    SUM((h.hours*h.discount*wt.price)+(u.amount*u.discount*i.purchaseprice))
    AS workPrice
RETURN c, inv, SUM(workPrice) as invoicePrice
```

```
                        Cypher with CALL (Task 4)

MATCH (inv:invoice) WHERE inv.customerId=0
CALL {WITH inv
  MATCH (c:customer)-[:PAYS]->(inv)
  RETURN c}
CALL {WITH c, inv
  MATCH (inv)-[:WORK INVOICE]->(w:work)
  RETURN w}
CALL {WITH w
  MATCH (wt:worktype)-[h:WORKHOURS]->(w)-[u:USED ITEM]->(i:item)
  RETURN SUM((h.hours*h.discount*wt.price)+(u.amount*u.discount*i.purchaseprice))
  AS workPrice}
RETURN c, inv, SUM(workPrice) AS invoicePrice
```

**Figure 6.** Queries for Task 4 (Invoice prices for a given customer).

```
                              SQL (Task 5)

WITH RECURSIVE sequential invoices AS
  (SELECT id, customerId, state, duedate, previousinvoice
   FROM invoice
   WHERE id=10000
UNION ALL
  SELECT i.id, i.customerId, i.state, i.duedate, i.previousinvoice
  FROM invoice AS i INNER JOIN sequential invoices AS j ON i.previousinvoice =
j.id
  WHERE i.previousinvoice <> i.id)
SELECT * FROM sequential invoices
```

```
                             Cypher (Task 5)

MATCH (i:invoice { invoiceId:10000 })-[p:PREVIOUS INVOICE *0..]->(j:invoice)
RETURN *
```

```
                        Cypher optimized (Task 5)

MATCH inv=(i:invoice { invoiceId:10000})-[p:PREVIOUS INVOICE *0..]->(j:invoice)
WHERE NOT (j)-[:PREVIOUS INVOICE]->()
RETURN nodes(inv)
```

**Figure 7.** Recursive queries for Task 5 (Invoices recursively related to a given invoice).

## 6. Test Executions

### 6.1. Test Settings

The tests were performed with a MacBook Pro Laptop with the following specifications:

- MacOS 12.3.1;
- 1.4 GHz quad core Intel Core i5;
- 8 GB 2133 MHz LPDDR3;
- Intel Iris Plus Graphics 645, 1536 MB.

MySQL versions 5.1.41 and 8.0.29, MariaDB version 10.8.3 and Neo4j community edition version 4.4.8 were installed on this computer. MariaDB and Neo4j were the latest versions at the time of the study. MySQL version 5.1.41 was already considered to be "end-of-life" when conducting the present study. However, it was used in a previous study [8] and version 5.1.42 was used in [7]. MariaDB driver version 2.7 and Neo4j driver version 4.1.1 were used. MySQL driver version 4.1.41 was used for MySQL 4.1.41.

### 6.2. Test Results

The results of tests are given in Tables 3 and 4. Each query result contains an average time for the query in milliseconds. Table 3 contains the results for the queries related to Tasks 1, 2, 3 and 4. Table 4 contains the result of recursive queries for Task 5. Table 4 does not contain results for MySQL 5.1 because MySQL 5.1 does not support those queries. The results are illustrated and further analyzed in the following subsections. Indexed (ind) is the same query on an indexed database. Notably, the performance ranking of different systems varied for different tasks and settings, with the exception that MySQL was always slower than MariaDB.

**Table 3.** Query performance of the MySQL, MariaDB and Neo4J.

|  | MySQL 5 | MySQL 8 | MariaDB | Neo4J | Neo4J CALL |
|---|---|---|---|---|---|
| Task 1 (Short Query) | | | | | |
| Avg | 576 | 464 | 486 | 162 | 149 |
| Avg, ind | 453 | 459 | 472 | 173 | 149 |
| Task 2 (Long Query) | | | | | |
| Avg | 6550 | 5337 | 5549 | 1868 | 1776 |
| Avg, ind | 5190 | 5257 | 5293 | 1968 | 1831 |
| Task 3 (Aggregate Query) | | | | | |
| Avg | 276,935 | 7674 | 7242 | 212,171 | 209,816 |
| Avg, ind | 251,138 | 7615 | 7117 | 215,053 | 205,198 |
| Task 4 (Aggregate Query with defined key) | | | | | |
| Avg | 3,938,500 | 5212 | 59 | 33 | 57 |
| Avg, ind | 3,891,082 | 5227 | 57 | 26 | 55 |

**Table 4.** Query performance in recursive queries.

|  | MySQL 8 | MariaDB | Neo4J | Neo4J Optimized |
|---|---|---|---|---|
| Recursive Query, 100 entities | | | | |
| Avg | 7850 | 9152 | 73 | 42 |
| Avg, ind | 1 | 1 | 72 | 42 |
| Recursive Query, 1000 entities | | | | |
| Avg | 79,037 | 92,917 | 331,338 | 2146 |
| Avg, ind | 2 | 4 | 208,573 | 2127 |

### 6.2.1. Query Performance in Task 1

The performance results for Task 1 are illustrated in Figure 8. From the generated dataset, the query returned 10,000 rows/objects. With this query, Neo4j outperformed relational databases. The best run of Neo4j was about three times faster than the best run on relational databases. Between MySQL 8.0 and MariaDB, no essential difference

appeared. Among versions of MySQL, the old version 5.1 was the fastest, when indices were used. In other databases, indexing played a minor role in efficiency. The inclusion of CALL brought minor benefits to Neo4j with this query. The numeric performance values are given in Table 3.

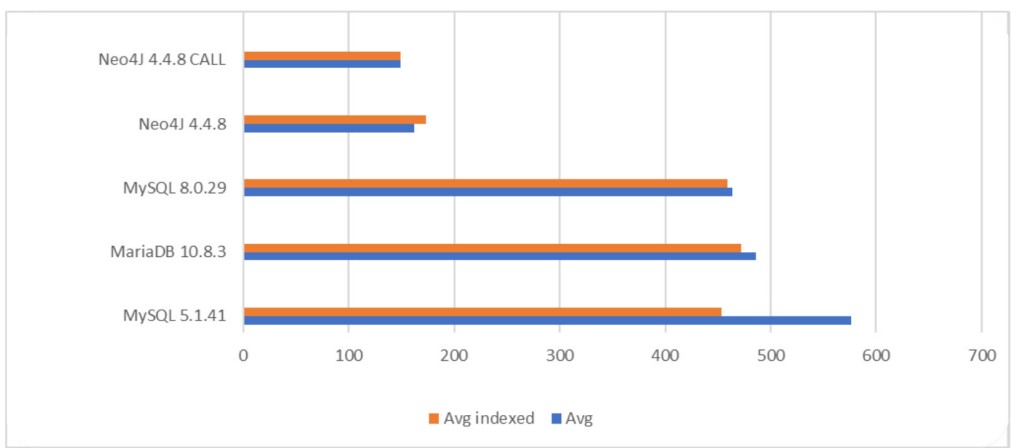

**Figure 8.** Results for the queries of Task 1.

6.2.2. Query Performance in Task 2

Results for the queries for Task 2 are shown in Figure 9. From the generated dataset, the query returned 10,000 rows/objects. The mutual difference between relational databases was similar to that in Task 1. The best run of Neo4J was about three times faster than the best run on relational databases. In this query, indices of relational databases had a minor effect. In basic Neo4j, indexing had a negative effect.

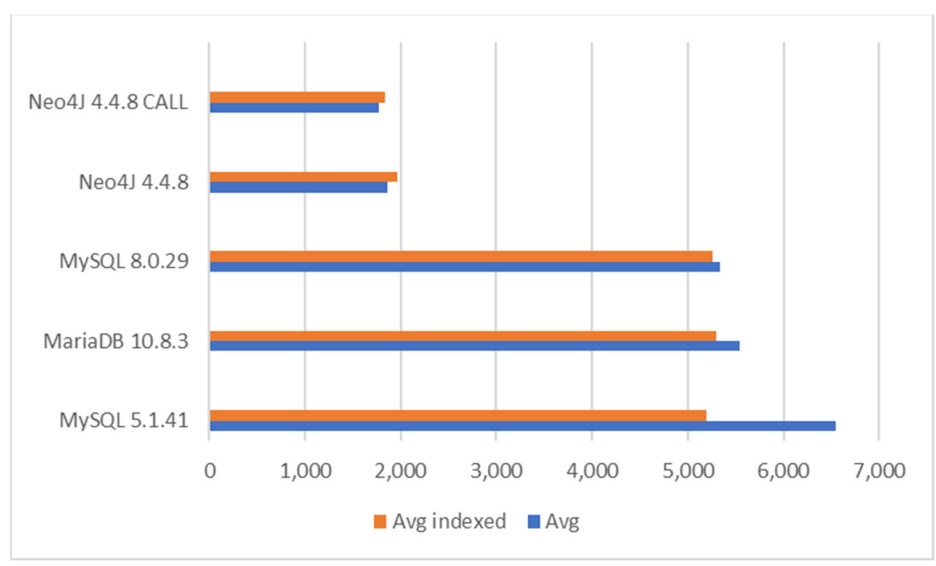

**Figure 9.** Results for the queries of Task 2.

6.2.3. Query Performance in Task 3

In the aggregation query of Task 3, radical differences between the database appeared. Here, MariaDB was 35 times faster than the old MySQL and 29 times faster than basic Neo4j. The inclusion of CALL did not provide a significant performance benefit compared to basic Neo4j. MariaDB was a little faster than MySQL 8.0.29. Figure 10 illustrates the differences between the databases and their settings.

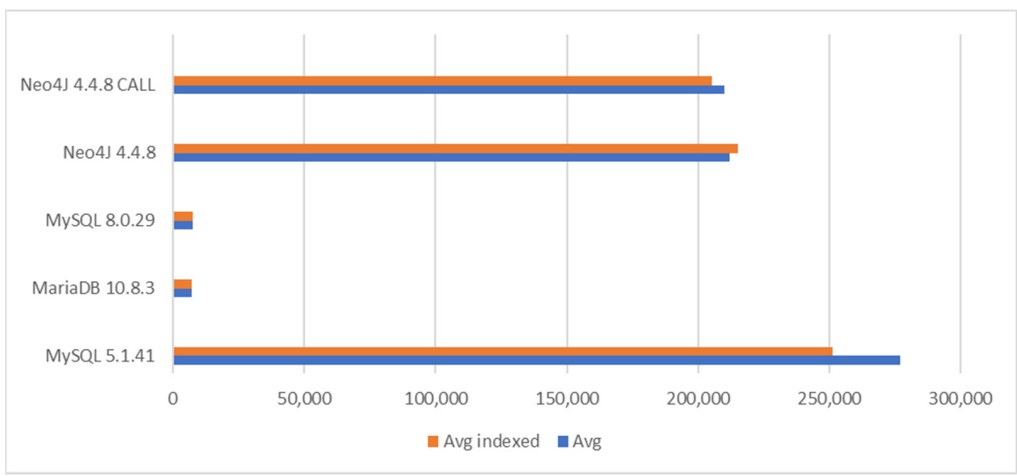

**Figure 10.** Results for the queries of Task 3.

6.2.4. Query Performance in Task 4

Results for Task 4 are illustrated in Figure 11. From the generated dataset, the query returned 10 rows/objects. MySQL 5.1.41 was excluded as the performance was too poor—the query took over one hour on average. In practice, it would be unusable. Neo4J performed the best. However, the inclusion of CALL did not lead to performance benefits. Instead, indexing seemed to bring improvements with the basic Cypher query. With indexing, Neo4j found the customer from the graph faster. MariaDB also performed well, but it was 50% slower than the best run with Neo4J. A dramatic difference appeared between relational database systems. Namely, MariaBD was about 90 times faster than MySQL 8.0.29. This was the first essential difference found between the modern databases.

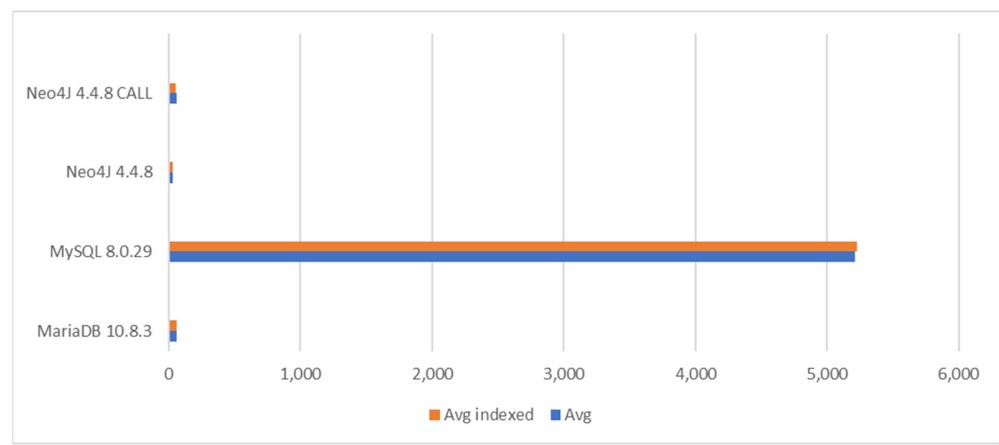

**Figure 11.** Results for the queries of Task 4.

6.2.5. Query Performance in Task 5

The recursive queries for Task 5 list all the sequential invoices related to the invoice with given id. The tests were performed with 100 and 1000 invoices. With 100 invoices, Neo4J outperformed relational databases when no indexing was used. When indices were used, the situation was reversed. Relational databases needed only one millisecond whereas Neo4J required 72 milliseconds and 42 milliseconds when indices were used. With 10,000 invoices and without indexing, Neo4'sj performance without optimization was poorest, but best with optimizations. Relational databases benefitted again from indexing considerably. Here, MYSQL 8.0.29 was the fastest and it was over 100,000 times faster than basic Neo4J and a thousand times faster than optimized Neo4J. MariaDB was five times slower than MySQL 8.0.29 but took only ten milliseconds. Moreover, this was

the second difference found between modern databases. Figure 12 represents the results when querying 100 sequential invoices and Figure 13 represents results when querying 1000 invoices.

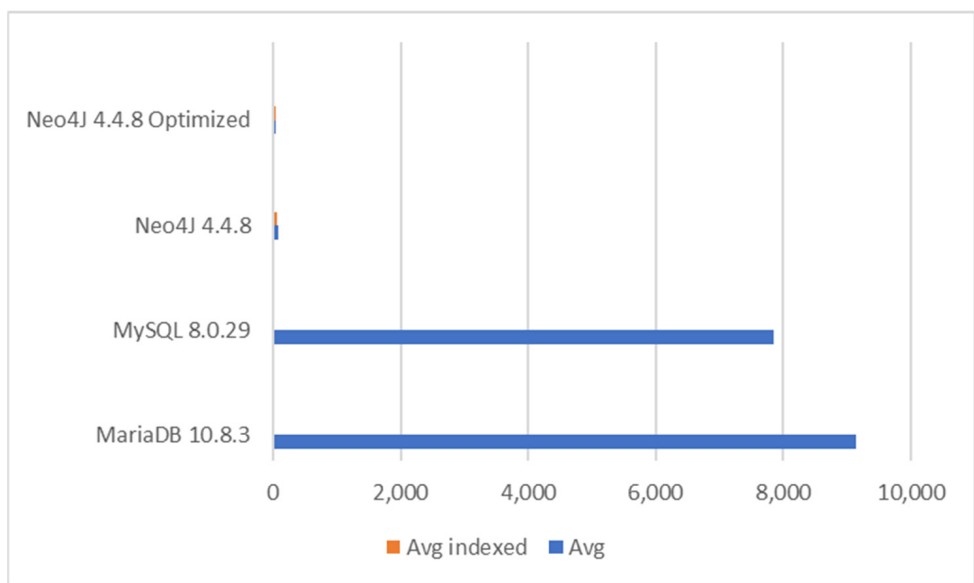

**Figure 12.** Results for the recursive queries among 100 entities.

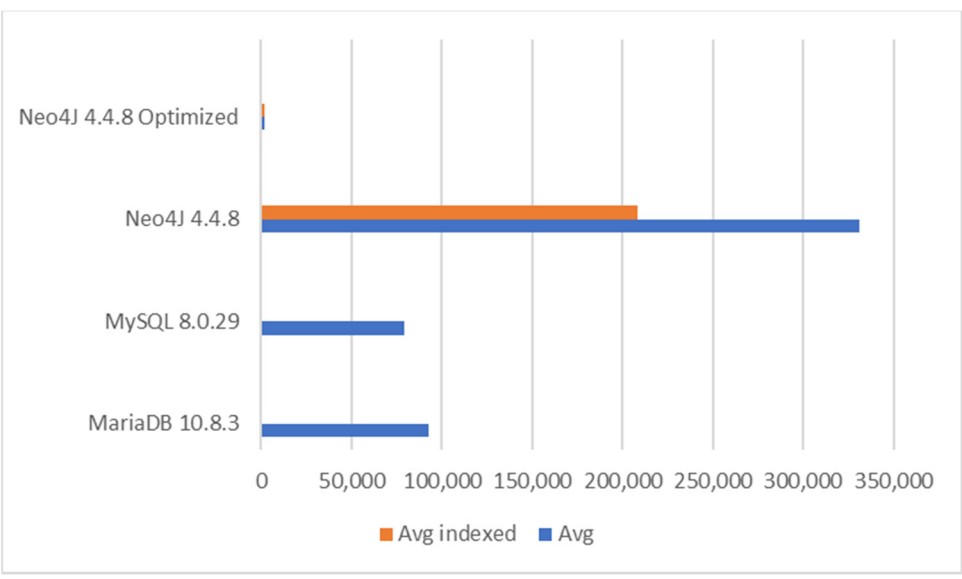

**Figure 13.** Results for the recursive queries among 1000 entities.

## 7. Discussion

In our tests, modern relational databases outperformed Neo4j in complex and recursive query tasks. When comparing Neo4j with MySQL and MariaDB, we also compared a Java program with a C/C++ program. Clearly, C/C++ has an upper hand because it does not run on a virtual machine. Additionally, relational database systems index primary keys and foreign keys by default. This provides a benefit in every query where tables are joined. Neo4j does not seem to benefit from indexing in many cases. One such case where indexing did benefit Neo4j was when Neo4j needed to find the starting point from the graph.

The benefit of indexing in MariaDB and MySQL is a benefit of the traditional relational database model. As the relations with the tables are created when executing the SQL query, indexing the keys becomes beneficial. The graph model does not benefit from such indexing

as there are no tables that are joined by keys. Querying a graph database is achieved by traversing the graph. One of the benefits of the graph model can be seen in recursive query tests. By optimizing the query, performance clearly improved, in this case, performing even better than an SQL database with Common Table Expressions. However, with recursive queries, indexing still brought dramatic benefits for the SQL database.

Compared with the modern databases, the old MySQL 5.1 performed well in simple query tasks. However, in aggregation tasks the performance of MySQL 5.1 collapsed. Further, MySQL 5.1 does not support recursive queries. MySQL 8.0.29 was more efficient in recursive queries than MariaDB with the long recursive query. MariaDB's efficiency in aggregate queries with a defined key was quite surprising. This may follow from the optimization algorithms of database management systems. Good optimization leads to performing the most selective operations first.

With the invoicing database schema used in the present study, the calculation of price was conducted with complex queries. If this database was used in a real case, the usage of table views would probably be preferred to simplify the queries. When it comes to using views, it is also of benefit that an SQL database outperformed a graph database, which is a new finding in this study that was not presented in previous studies. In previous studies, Neo4j typically outperformed SQL databases. In study [4] for example, Neo4j outperformed Oracle in various tests using count(*) queries. In the present study, aggregation queries were also used but the result was different. The present study also indicated the benefit of indexing in SQL databases in many of the tests. SQL databases seemed to benefit from indexing and in some cases very dramatically. However, Neo4j did not seem to benefit from indexing, apart from when a starting point in the graph was indexed.

In further studies, it is essential to compare other NoSQL database systems to modern relational database systems. Nowadays, the general understanding is that NoSQL database systems are more efficient than relational database systems in general. It is evident that in performance studies, indexing, optimization and query complexity should be taken into account as was the case in the present study.

## 8. Conclusions

The present study compared relational database systems (MariaDB and two versions of MySQL) and a graph database system (Neo4j) efficiency using queries with different complexities. The results support earlier studies where graph database systems outperformed relational database systems with structurally simple datasets and simple queries. However, with more complex queries new relational database systems outperformed Neo4j.

The significantly better performance of new relational database systems compared to MySQL 5.1 is not surprising as the tested MariaDB and MySQL 8.0.29 versions are 10 years newer, and many developments have occurred during that time. Although MariaDB is based on old MySQL, it offers a different feature set and is completely open source [25]. One significant change after MySQL 5.1.41 is a change in the default storage engine from MyISAM to InnoDB in version 5.5 [26]. InnoDB is used as a default storage engine of MariaDB. The study indicates the extent to which relational database query performance has improved during the last one and half decade.

Neo4j outperformed modern relational database systems in most of the query tasks. Using the best settings of database systems, Neo4J was often at least three times faster than modern relational databases. However, in the task where an aggregated value was calculated for the given entity, Neo4J was 200 times faster than MySQL 8.0.29. In this task, the most essential difference between modern databases also appeared. MariaDB was over 90 times faster than MySQL 8.0.29. In the most complex query task, MariaDB was 29 times faster than Neo4j when indices were used and Neo4J query was optimized. In the same setting, MySQL 8.0.29 was 27 times faster than Neo4J. The role of optimization and indexing played an essential role in performance, especially in the long recursive query. Without indexing, basic Neo4J was the slowest, but the optimized query was the fastest. Indexing changes the situation, i.e., relational database systems outperformed Neo4J. MySQL 8.0.29

performed best. It was over 1000 times faster than the optimized Neo4J query and over 100,000 times faster than basic Neo4J.

Our general conclusion is that on the basis of tests with our data set and queries, it cannot be generally concluded which of the database systems possesses the best query efficiency. In other words, the efficiency depends on the complexity of data and queries. Furthermore, query optimization and indexing may play important roles. This means that when choosing a database for an application domain, the query needs must be analyzed carefully beforehand. The results in the present study show how a relational database system is still a good alternative when it comes to performance compared with an NoSQL graph database.

**Author Contributions:** Conceptualization, P.K., M.J. and J.N.; Methodology, P.K., M.J. and J.N.; Software, P.K.; Validation, P.K., M.J. and J.N.; Formal Analysis, P.K., M.J. and J.N.; Investigation, P.K. and M.J.; Data Curation, P.K.; Writing—Original Draft Preparation, P.K., M.J. and J.N.; Writing—Review and Editing, P.K., M.J. and J.N.; Visualization, P.K. and M.J. All authors have read and agreed to the published version of the manuscript.

**Funding:** This research received no external funding.

**Institutional Review Board Statement:** Not applicable.

**Informed Consent Statement:** Not applicable.

**Data Availability Statement:** The source code for generating the data is available in GitHub: https://github.com/homebeach/InvoicingDBTestBench (accessed on 2 January 2022).

**Conflicts of Interest:** The authors declare no conflict of interest.

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
