# Peer review of "Performance of Graph and Relational Databases in Complex Queries"

_applsci, doi:10.3390/app12136490_

Round 1
Reviewer 1 Report
Authors aims to present a comparison of the query performance of a graph-based database (Neo4j) and relational databases (MySQL and MariaDB). They try to evaluate the effect of different efficiency issues like indexing and optimization.
Some details should be given to the choices made, namely DBMS used and, most important, why the setup illustrates "complex queries".
The authors may find below a list of suggestions for improvement.
1) In abstract (and further in the Introduction) it is not well explained why the graph databases possess better performance than relational databases. In which circunstancies? The paradigms are quite distinct and both types have good performance in specific scenarios. It is not possible to compare incomparable realities.
2) Regarding the test scenario, it shows a typical application of an enterprise information system. The proposed scenario is in fact a "large dataset" as argued by the authors?
What could be the results in a Bioinformatics scenario, a typical "big data" with a high volume of transactions?
3) Some confusion about the use of both MariaDB and MySQL. Why to have "an old MySQL version"? The argue of "making our research compatible with earlier studies" is a bit weak, as the "earlier studies" do not use the same scenario. I'd say that a comparison between Neo4J and MySQL or MariaDB would be enough. BTW MariaDB and MySQL are from the same family, being the latter more recent.
4) What is a "complex query"? Details are expected, to identify the relevance of the study being conducted.
5) Does "views" have impact in the performance observed in relational databases?
6) Table 1 (pp.6) - Table "Customer" has an index column (ID), right? If not, why?
7) Suggestion: tables 2 to 6 should be better as figures.
8) With the inputs provided in Table 7, how many records were collected by each SQL/Cypher instruction?
9) Tables 8 and 9 (pp.11 and 12) - what means these values? milliseconds?
10) bibliography: too outdated. More recent entries should be included.
Summing up, the work has merit, however I am unconfident about the "complexity" of these queries and the narrow volume of records in each table.
More details should be given regarding the motivation behind this work and the really meaning of "complexity" in this context.
Author Response
The authors may find below a list of suggestions for improvement.
1) In abstract (and further in the Introduction) it is not well explained why the graph databases possess better performance than relational databases. In which circunstancies? The paradigms are quite distinct and both types have good performance in specific scenarios. It is not possible to compare incomparable realities.
AUTHORS’ REPLY: This is introduced now more detailed:
The graph database is a NoSQL paradigm where the navigation is based on links instead of joining tables. Links can be implemented as pointers, and following pointers is a constant time operation, whereas joining tables is more complicated, even in the presence of foreign keys. Therefore, link-based navigation has been seen as a more efficient query approach than using join operations on tables.
2) Regarding the test scenario, it shows a typical application of an enterprise information system. The proposed scenario is in fact a "large dataset" as argued by the authors?
AUTHORS’ REPLY: Considering that some of the runtimes are already infeasible, we believe that the datasets can be called large for these queries.
What could be the results in a Bioinformatics scenario, a typical "big data" with a high volume of transactions?
AUTHORS’ REPLY: This would be very interesting further research. We need suitable test data and typical use cases. We thank the reviewer for this idea, which we will consider in the future.
3) Some confusion about the use of both MariaDB and MySQL. Why to have "an old MySQL version"? The argue of "making our research compatible with earlier studies" is a bit weak, as the "earlier studies" do not use the same scenario. I'd say that a comparison between Neo4J and MySQL or MariaDB would be enough. BTW MariaDB and MySQL are from the same family, being the latter more recent.
AUTHORS’ REPLY: We used the old MySQL to make our research compatible with an earlier study where the old version was used. The revised version contains new MySQL. Our environment has been changed and we had to run all the queries in the same environment. We also used the latest version of Neo4J. For these reasons, all run times have changed. The conclusions follow our new results.
4) What is a "complex query"? Details are expected, to identify the relevance of the study being conducted.
AUTHORS’ REPLY: This is clarified in the revised manuscript as follows: ‘In a complex query, the necessary data must be collected from several tables in a SQL database, or a by traversing a path of different types of nodes, potentially using recursion, in a graph database. Using a complex query an aggregated value (e.g., a count or an average) from a large data set be calculated.’
5) Does "views" have impact in the performance observed in relational databases?
AUTHORS’ REPLY: Views are just prewritten parts of a query, and they have no relation to performance issues.
6) Table 1 (pp.6) - Table "Customer" has an index column (ID), right? If not, why?
AUTHORS’ REPLY: The ID of the customer table is indexed because every primary key is automatically indexed in relational databases. Table 1 represents additional indices. We have clarified this.
7) Suggestion: tables 2 to 6 should be better as figures.
AUTHORS’ REPLY: They are figures now.
8) With the inputs provided in Table 7, how many records were collected by each SQL/Cypher instruction?
AUTHORS’ REPLY: A row is a data record in relational databases and an object is a data record in graph databases. In other words, their number corresponds to the number of records.
9) Tables 8 and 9 (pp.11 and 12) - what means these values? milliseconds?
AUTHORS’ REPLY: This is explained in Section 6.2: “Each query result contains an average time for the query in milliseconds.”
10) bibliography: too outdated. More recent entries should be included.
AUTHORS’ REPLY: The bibliography is updated. We found three new relevant references.
Summing up, the work has merit, however I am unconfident about the "complexity" of these queries and the narrow volume of records in each table.
More details should be given regarding the motivation behind this work and the really meaning of "complexity" in this context.
AUTHORS’ REPLY: Of course, the data set could be larger, but we think that it is enough large for testing the performance of databases. The performance differences of databases clearly appear by the used data set. The slowest single runs take more than one hour. Running the whole test set takes a few days.
AUTHORS’ GENERAL REPLY: We hope that the comments and the revised manuscript give answers to these issues. We thank the reviewer on good comments that helped us to improve the manuscript.
Reviewer 2 Report
In this paper the authors compare the query performance of a graph-based database (Neo4j) with the query performance of two popular RDBMSs. MySQL and MariaDB.
1. Although many similar studies exist in the literature, this one also takes into account queries of higher complexity in the conducted experiments. The results of these experiments are somehow surprising, since RDBMSs seem to outperform the noSQL Neo4j graph database when processing complex queries.
This conclusion contradicts the findings of other similar studies that report performance gaps of more than one order of magnitude (in favor of Neo4j). However, one of the most important advantages of this work is that the authors have published their code on GitHub for reproducibility reasons.
2. The related works section is well-written. The relevant articles are appropriately discussed and their key findings are conveniently summarized in each paragraph of this section.
3. There is a considerable number of syntactical and grammatical errors in the article. There many articles (the, a) that are missing from the text. For example,
• in line 27, “In the present study, we compare traditional relational model” should be “In the present study, we compare the traditional relational model”.
• Line 34: “have been compared” -> “was compared”
• Line 98” “factors 5 and 7-9” -> “factors of 5 and 7-9”
• Figure 1: Table label “Targer” should have been “Target”
4. In line 48 the authors mention that “MariaDB is a modern version of MySQL”. This is not accurate; both systems are developed and updated individually. Both of them are quite modern. They share a common past, since MariaDB was forked from MySQL. Please rephrase.
The same error is repeated in line 75: “MariaDB is a descendant of MySQL”. MySQL is still supported and frequently updated. Although they share a common past, they are both modern systems that evolve in parallel.
5. In line 76: what does this ranking represent? Does it rank the database systems by popularity? Please clarify.
6. The schema of Figure 1 looks counter-intuitive to me. In my opinion, the WorkInvoice and WorkTarget tables could have been removed completely.
The Work table can be directly connected to the Invoice table and the Target table by inserting two fields (set as Foreign Keys): InvoiceID and TargetID.
The authors must describe their example more clearly, because in the eyes of the reader this schema includes redundant elements.
7. In Section 4, the authors describe how the databases are filled with data, but they do not provide an insight of the generated data volumes (i.e. number of records, data sizes in MB/GB, etc.). Table 7 is presented later in the article. In my opinion it must be moved to Section 4.
8. The MySQL version used in the experiments is ancient. MySQL 5.1 was released in 2008 and discontinued at 2013. The current MySQL version is 8.0.29 and includes several optimizations compared to the previous versions.
In line 284, the authors mention that they use this version for comparison purposes. However, MySQL 8.0 must be also included in the experiments, along with MySQL 5.1. Otherwise, the presented results are not reliable and untrustworthy. So the conclusions from this work will not be safe.
Author Response
In this paper the authors compare the query performance of a graph-based database (Neo4j) with the query performance of two popular RDBMSs. MySQL and MariaDB.
1. Although many similar studies exist in the literature, this one also takes into account queries of higher complexity in the conducted experiments. The results of these experiments are somehow surprising, since RDBMSs seem to outperform the noSQL Neo4j graph database when processing complex queries.
This conclusion contradicts the findings of other similar studies that report performance gaps of more than one order of magnitude (in favor of Neo4j). However, one of the most important advantages of this work is that the authors have published their code on GitHub for reproducibility reasons.
AUTHORS’ REPLY: Thank you!
- The related works section is well-written. The relevant articles are appropriately discussed and their key findings are conveniently summarized in each paragraph of this section.
AUTHORS’ REPLY: Thank you!
3. There is a considerable number of syntactical and grammatical errors in the article. There many articles (the, a) that are missing from the text. For example,
• in line 27, “In the present study, we compare traditional relational model” should be “In the present study, we compare the traditional relational model”.
• Line 34: “have been compared” -> “was compared”
• Line 98” “factors 5 and 7-9” -> “factors of 5 and 7-9”
• Figure 1: Table label “Targer” should have been “Target”
AUTHORS’ REPLY: These are corrected and the language polished in general.
4. In line 48 the authors mention that “MariaDB is a modern version of MySQL”. This is not accurate; both systems are developed and updated individually. Both of them are quite modern. They share a common past, since MariaDB was forked from MySQL. Please rephrase.
The same error is repeated in line 75: “MariaDB is a descendant of MySQL”. MySQL is still supported and frequently updated. Although they share a common past, they are both modern systems that evolve in parallel.
AUTHORS’ REPLY: We agree with the reviewer. Our conceptualization was improper. We meant that MariaDB is a descendant of MySQL 5. However, these issues are rewritten following reviewer’s instructions.
- In line 76: what does this ranking represent? Does it rank the database systems by popularity? Please clarify.
AUTHORS’ REPLY: Yes, clarified.
6. The schema of Figure 1 looks counter-intuitive to me. In my opinion, the WorkInvoice and WorkTarget tables could have been removed completely.
The Work table can be directly connected to the Invoice table and the Target table by inserting two fields (set as Foreign Keys): InvoiceID and TargetID.
The authors must describe their example more clearly, because in the eyes of the reader this schema includes redundant elements.
AUTHORS’ REPLY: In the manuscript we say: ‘Relationships between the entities are stored in relationship tables worktarget, workinvoice, useditem and workhours. These represent many-to-many relationships between entities.’
- In Section 4, the authors describe how the databases are filled with data, but they do not provide an insight of the generated data volumes (i.e. number of records, data sizes in MB/GB, etc.). Table 7 is presented later in the article. In my opinion it must be moved to Section 4.
AUTHORS’ REPLY: A row is a data record in relational databases and an object is a data record in graph databases. In other words, their number corresponds to the number of records. The sizes of the databases are inserted. Moving that table is a good idea.
The MySQL version used in the experiments is ancient. MySQL 5.1 was released in 2008 and discontinued at 2013. The current MySQL version is 8.0.29 and includes several optimizations compared to the previous versions.
In line 284, the authors mention that they use this version for comparison purposes. However, MySQL 8.0 must be also included in the experiments, along with MySQL 5.1. Otherwise, the presented results are not reliable and untrustworthy. So the conclusions from this work will not be safe.
AUTHORS’ REPLY: We understand the reviewer’s point. MySQL 8.0 is now added into the study. Our environment has been changed and we had to run all the queries in the same environment. We also used the latest version of Neo4J. For these reasons all run times have changed. The conclusions follow new results.
AUTHORS’ GENERAL REPLY: We thank the reviewer good comments that helped us to improve the paper.
Reviewer 3 Report
the paper should clearly indicate the novelty in abstract as well as introduction, discussion and conclusion
The paper suggests to work on dramatically the efficiency of relational database but whereas new concept and modules has overlaid the present concept, A paper will be more efficient where it can have a comparitive overview in the past with there recent model discussed in form of table to be included in methodology
Author Response
AUTHORS’ REPLY: The manuscript has been improved at all levels following the reviewers’ comments. A reviewer required that MySQL 8.0 had to be included in comparison. Our environment has been changed and we had to run all the queries in the same environment. We also used the latest version of Neo4J. For these reasons all run times have changed. The conclusions follow new results.
Round 2
Reviewer 1 Report
The paper was improved in this second version and the changes made answer to my comments.
Reviewer 2 Report
The quality of this article was substantially improved in the revised version. All my comments have been sufficiently addressed. I am most happy with the inclusion of MySQL 8.0 in the experiments.
Consequently, in my opinion the article is now ready for publication.